# Unraveling the dynamics of dengue in Metahara town, East Shewa, Oromia, Ethiopia, 2023

**Bikila Negesa Gobena** **[1]\*, Teshome Kabeta Dadi[2], Gemechu Chemeda Feyisa[2], Birhanu Kenate[3], Gemechu Shumi[3], Fantahun Workie[4], Haimanot Workie[4], Ebise Djirata[4], Dabesa Gobena[3,5]**

**1** Ethiopian Field Epidemiology and Laboratory Training Program Resident, Jimma University, Jimma, Ethiopia, **2** Department of Epidemiology, Faculty of public health, Jimma University, Jimma, Ethiopia, **3** Public Health Emergency Management and Health Research Directorate, Oromia Health Bureau, Addis Ababa, Ethiopia, **4** Disease and Health Events Surveillance and Response Directorate, Ethiopian Public Health Institute, Addis Ababa, Ethiopia, **5** School of medical laboratory science, Institute of Health, Jimma University, Jimma, Ethiopia

\* sichanket2018@gmail.com

## Abstract

### Background

Since 2013, dengue cases have shown a marked increase in Ethiopia. The current suspected outbreak occurring in Metahara town, Oromia Regional State, began in July 2023. This study aimed to confirm and characterize the outbreak, identify risk factors, and implement control measures.

### Methods

We conducted a descriptive study and an unmatched case-control design, using a one-to-two ratio of cases to controls. We collected data on the dengue outbreak using line lists, laboratory test results, environmental observations, home visits, and entomological examinations. We selected a total of 50 cases using simple random sampling from the line list and purposively chose 100 controls from the same block. We applied community-based face-to-face interviews with 150 participants. After gathering data through Kobo Collect, we analyzed it using Statistical Package for the Social Sciences (SPSS) version 26 and summarized the findings in Microsoft Excel 2013. A binary logistic regression model was employed to identify significant variables, with p-values ≤ 0.25 in bivariate analysis considered for the final model. Crude and adjusted odds ratios (OR and AOR) were used to measure associations, with p-values ≤ 0.05 indicating significance.

### Results

The investigation confirmed 342 dengue cases, corresponding to an attack rate of 7.1 per 1,000 population and a case fatality rate of 0.88%. Significant risk factors included not using long-lasting insecticide nets during the daytime (9-fold increased likelihood) and having open water containers (5-fold increased likelihood. Respondents lacking disease

**Data availability statement:** All relevant data are in the manuscript and its Supporting Information files.

**Funding:** The author(s) received no specific funding for this work.

**Competing interests:** The authors have declared that no competing interests exist.

awareness were 25 times more likely to be infected, while wearing long-sleeved clothing conferred a protective effect of 75% reduction in risk.

## Conclusion

The dengue outbreak in Metahara town was driven by epidemiological, entomological, and environmental factors, with Aedes aegypti as the primary vector. The ongoing circulation of DENV-3, coupled with insufficient vector control measures, poses a serious public health threat. Key contributing factors to the outbreak include the lack of utilization of long-lasting insecticide nets (LLINs) during the daytime, improper water storage practices, insufficient public knowledge regarding transmission and prevention strategies, and inadequate protective clothing choices that increase vulnerability to mosquito bites. Strategies including vector control, community education, promotion of protective clothing, and improved surveillance were recommended.

## Author summary

In early 2023, the Town of Metahara experienced a significant outbreak of dengue. The authors conducted an investigation into how the disease spread, what factors enhanced the outbreak, and the strategy to manage the dengue outbreak. Accordingly, the outbreak was by *Aedes aegypti* mosquitoes, the primary vector for dengue transmission. Intermittent rains, inappropriate water storage practices, and numerous old tires in town before the outbreak created ideal breeding grounds for these mosquitoes. Additionally, the author identified poor sanitation, lack of vector control measures, and limited public awareness as contributing factors. The outbreak escalated quickly, with hundreds of cases reported between July 25 and September 5, 2023. Demographic and geographic distribution of cases and identified neighborhoods and vulnerable population that were disproportionately affected had been assessed. The author's comprehensive approach, such as active surveillance and case reporting, vector control activities, improved access to diagnostic testing and clinical management and launching public awareness campaigns, provided valuable insights into outbreak control. The authors' investigation provides the challenges faced by the health system in managing the outbreak, such as limited resources, infrastructure constraints, and coordination issues between different agencies. The author recommended the town's preparedness and response to future dengue outbreaks. These include improving vector surveillance and control, investing in healthcare infrastructure, and enhancing community engagement and education programs. The thorough investigation carried out by the author sheds light on the dynamics of the dengue outbreak in Metahara town and provides valuable recommendations for enhancing the management of similar public health emergencies.

## Introduction

Dengue, a global public health concern caused by the mosquito-borne virus, affects 390 million people annually in tropical and subtropical regions [1]. Its symptoms can range from mild flu-like symptoms to severe complications which can be life-threatening, even death [2,3]. Since 2013, dengue outbreaks have increasingly burdened Ethiopia's healthcare system [4]. The detection of anti-DENV IgG/IgM antibodies in feverish patients helped identify

circulating dengue infections across the country [5,6]. In 2017 and 2019, it emerged in the Afar region of Ethiopia, affecting more than 100 individuals [7,8].

Previous studies reveal variations in dengue risk factors, including past infection, social and environmental factors, knowledge, and sustainable vector control operations [4–6,8,9]. On the other hand, daytime utilization of Long-Lasting Insecticide Nets (LLINs) has protected against dengue transmission, and the spatial analysis showed a significant decrease in control houses within 50 meters of bed-net houses one month after the intervention [10]. However, its application is limited in Sub-Saharan Africa, including Ethiopia [11].

Close contact play a crucial role in study outcomes since being near dengue-infected individuals increases the risk of transmission through mosquito bites. Asymptomatic infections often occur among those in close community ties, highlighting the importance of understanding how transmission dynamics work within these groups [12]. Educational status significantly influences individuals' understanding and practices of dengue prevention. Higher education levels correlate with better knowledge of preventive measures like using Long-Lasting Insecticide Nets (LLINs) and removing mosquito breeding sites. Well-informed individuals are more likely to adopt effective prevention strategies [13]

Increased incidence of dengue is strongly correlated with high temperatures, humidity, and prolonged rains. Other notable risk factors for dengue infection include those that encourage mosquito breeding, such as having exposed, stagnant water in containers (like buckets, drums, tires, pots, etc.), and those that increase vector-human contact, such as the absence of door or window screens and the non-use of personal protective measures (like repellents) and travel history within or between countries endemic for dengue. Vector control, through chemical or biological targeting of mosquitoes and removal of their breeding sites, is the mainstay of dengue prevention [8,14,15].

To effectively combat dengue, public health authorities should adopt an integrated vector management (IVM) approach. This strategy combines chemical control using insecticides, biological control through natural predators, and environmental management to eliminate mosquito breeding sites [16]. Community involvement is vital for the success of these interventions, while robust surveillance systems help track mosquito populations and guide targeted actions. By addressing multiple aspects of vector control, IVM reduces dengue transmission and promotes sustainable practices, lowering the risk of insecticide resistance and improving public health outcomes [17]

Ethiopia has enhanced its disease surveillance through the Integrated Disease Surveillance and Response (IDSR) strategy, improving data collection and outbreak response. The Ethiopian Public Health Institute (EPHI) has implemented a multi-hazard approach since 2009, including community and event-based surveillance systems for early detection of infectious diseases [18,19].The country has received less attention in disease surveillance for arboviral diseases like yellow fever, dengue, Chikungunya, and sand fly fever, despite their high prevalence in neighboring countries such as Sudan, Eritrea, Kenya, and Djibouti [20].

While dengue is vaccine-preventable, vaccines like *Dengvaxia* and *Qdenga* are not yet part of Ethiopia's immunization program. The recent WHO prequalification of *Qdenga* may facilitate future availability [21,22].

From the above points of view, the burden of dengue in Ethiopia is significantly underreported. To the best of my knowledge, in Metahara Town, which is located in the Oromia Regional State, there is no data about the confirmed cases and associated factors of dengue. This study aimed to investigate the outbreak, determine its associated factors, and undertake appropriate public health control measures for the dengue outbreak..

This manuscript details the magnitude, clinical aspects, and factors related to the dengue outbreak in Metahara, Ethiopia, offering valuable insights for future preparedness and mitigation strategies. This study was part of the Ethiopian Field Epidemiology and Laboratory Training Program (EFELTP), which is a nationwide initiative aimed at providing residents

with practical opportunities to apply epidemiological concepts to public health issues, as well as equipping them with essential research techniques and an understanding of disease patterns and risk factors.

## Method and materials

### Ethics statement

The Ethical Review Committee of the Oromia Health Bureau granted approval for this study (Reference Number: BFO/HBTFH/I/G/10059) on September 3, 2023. Given the urgent public health crisis during the dengue outbreak investigation, we obtained verbal consent to promptly engage participants, also considering the guardians when the participant was a child. We documented the verbal consent to ensure transparency and accountability. To maintain confidentiality, we assigned unique codes and conducted data analysis anonymously.

### Study area and period

Metahara Town, in the East Shewa Zone of Oromia, Ethiopia, lies 193 km from Addis Ababa and experiences a hot, partly cloudy climate with annual temperatures ranging from 67°F to 93°F. According to the town administrative report for 2022/23, Metahara has a population of 47,661, with one health center, two health posts, and 13 private clinics [23,24]. The map of Metahara Town was developed in QGIS using freely available shapefiles. The process involved importing the original data, refining the geometries, and updating the attribute information. These modifications, aligned with standard GIS practices, ensured that the map was accurate and valuable for public health analysis. (See Fig 1)

### The study design

We conducted a descriptive study and an unmatched case-control design, using a one-to-two ratio of cases to controls.

### Source population

All residents of Metahara town.

### Study population

All randomly selected patients having a positive laboratory test result for DENV using RT-PCR or RDT or epidemiologically linked cases during the study period. A control was defined as any person living in the same block as a case, but no signs or symptoms of the disease during the study period.

### Sample size

We calculated the sample size using EPi-Info Version 7, setting the power at 80%, the confidence interval (CI) at 95%, the exposure rate among controls at 80.7%, and the adjusted odds ratio (AOR) at 0.32. We established a case-to-control ratio of 1:2, resulting in a total of 150 study participants [25].

### Sampling procedure

We developed the sampling frame based on a line list of dengue patients. To select study participants, we utilized a simple random sampling technique. For every confirmed case of dengue, we randomly chose two individuals without dengue from the same block to act as controls in the study.

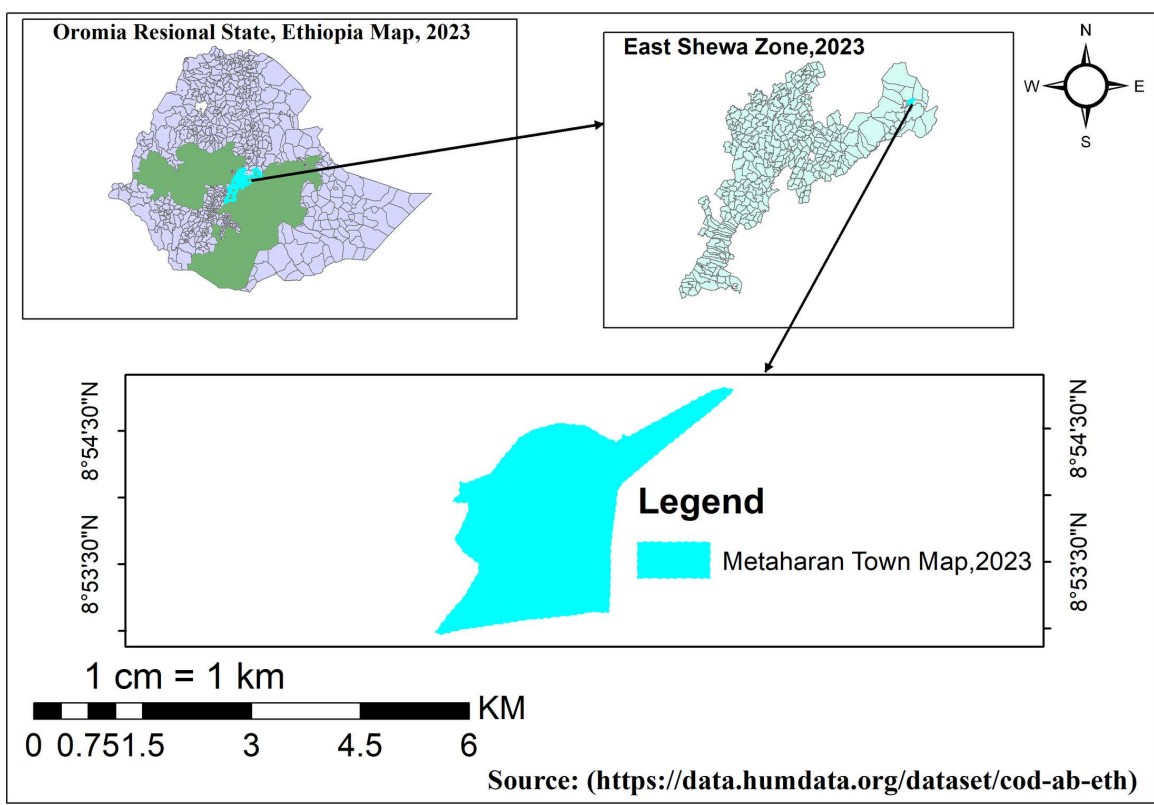

**Fig 1. Map of Metahara Town, East Shewa, Oromia Regional State, Ethiopia, 2023 (https://data.humdata.org/dataset/cod-ab-eth).** We conducted from September 7 to 30, 2023.

## Laboratory result interpretation

Following the notification of a dengue case, eleven samples were collected from 11 patients using aseptic techniques, with each patient providing 4 mL of serum during the acute phase (0–7 days post-symptom onset. The samples were placed in serum separation tubes, centrifuged to prevent hydrolysis, and sent to the Ethiopian Public Health Institute the only national laboratory capable of performing RT-PCR tests for arboviral diseases while adhering to all applicable regulations for sample transport and storage at -20°C.

Nucleic acid detection assays consist of three main steps: extraction and purification of nucleic acids, amplification, and detection and characterization of the amplified products. Out of the 11 samples tested, four were positive for DENV-3 and classified as index cases. The STANDARD Q Dengue NS1 Ag test demonstrates strong validity, with a sensitivity of 92.9% and specificity of 98.7%, effectively detecting dengue virus infections in serum, plasma, or whole blood samples.

We initially collected 100 mL of whole blood from suspected cases using a micropipette and added it to the sample well of the test device. After a 20-minute waiting period, results were interpreted: a positive result for IgM antibodies was indicated by red lines appearing in both the control and NS1 sections, while a negative result showed only the control line in red. If no red lines appeared in either section, or if there was no red line in the test section while the control line was absent, the result was deemed invalid.

### The eligibility criteria

We included all confirmed and epidemiologically linked dengue cases from both public and private healthcare facilities between August 3 and September 5, 2023, as confirmed cases. For our control group, we selected individuals from Metahara town who resided in the same block as a confirmed case but did not show any signs or symptoms of dengue. We excluded seriously ill suspected cases and non-residents of Metahara.

### Data collection procedures

We carried out data collection through face-to-face interviews, document reviews, and the examination of a line list of patients, complemented by environmental investigations conducted by four diploma nurses. To facilitate effective data gathering, we gave training on the Kobo Toolbox and the structured data collection tool. We assessed sanitation practices, the presence of stagnant water, methods of water collection, drainage systems, and individual mosquito protection measures. We collected primary data using face-to-face interviews through the Kobo Collect mobile app. The structured questionnaires adapted from previous studies have been used [9,26]. The study included four parts:

**Socio-demographic characteristics.** This section covered information such as sex, age, ethnicity, education, and income.

**Clinical presentation.** This part focused on the symptoms and signs of dengue

**Risk factors.** Here, we looked at factors like the availability and use of Long-Lasting Insecticide Nets (LLINs), water containers in and around the home, travel history, and previous dengue infections.

**Knowledge of dengue.** This section assessed participants' understanding of dengue fever.

We focused on children under five and adults with speaking or hearing difficulties during data collection. Caregivers answered for young children, and translators used body language for those facing communication barriers.

### Definition of terms

**An epidemiological case** was defined as any individual residing in Metahara town between July 25 and September 5, 2023, who exhibited a fever lasting 2 to 7 days. In addition to the fever, these individuals experienced at least one of the following symptoms: headache, joint pain (arthralgia), muscle pain (myalgia), rash, or bleeding from any part of the body. Furthermore, they tested negative for malaria or shown a positive result for IgM antibodies to be classified as an epidemiologically linked.

**A confirmed case** was defined as one that was classified as confirmed following a positive result for DENV from an RT-PCR laboratory test. In contrast, a control was any individual residing in the same block as a patient identified with a dengue case, but who exhibits no signs or symptoms of the disease. This definition applies to the period from July 25 to September 5, 2023.

**Close contact** was defined as being within 1 meter of a confirmed dengue case for at least 15 minutes during the two-week period from July 25 to September 5, 2023.

**No close contact** means that an individual had not been within the specified proximity of a confirmed case during the timeframe from July 25 to September 5, 2023.

### Data processing and analysis

The data relevant to the dengue outbreak, including socio-demographic factors, geographical distribution, clinical presentation, and the total number of cases, was taken from a line list

and laboratory test results through document review. The number of breeding sites and larvae sources were collected using environmental observation, home-to-home visits, and entomological examination. Community-based face-to-face interviews of patients who had a history of dengue and those without a history of dengue after the outbreak stopped were collected using the Kobo Collect mobile app. We checked data for completeness, coded, and cleaned. The 2023 population of the Metahara town administrative report was used to calculate the magnitude of the dengue outbreak, and the data in the Kobo Toolbox was exported to SPSS version 26 for analysis. We used numbers and percentages to summarize categorical variables, while continuous variables were summarized as mean ± SD. We used binary logistic regression to analyze the data and determine associations. All variables with a p-value of ≤ 0.25 and significant in the chi-square test were candidates for multivariate analysis. We used the crude odds ratio (COR) and adjusted odds ratio (AOR) to measure the strength of the association. In the multivariate analysis, all variables with a p-value ≤ 0.05, along with the adjusted odds ratio (AOR) and 95% confidence interval (CI), were declared independent variables that were statistically associated with dengue outbreak. The Hosmer-Lemeshow test was used to assess the model's goodness of fit.

## Results

### Descriptive part

**Dengue outbreak verification.** Metahara Health Center alerted surveillance officers to a rise in cases of fever, headache, joint pain, nausea, and vomiting, all of which tested negative for malaria. In response, the Ethiopian Public Health Institute (EPHI) and the Oromia Health Bureau (OHB) deployed a team. Eleven suspected dengue cases were sampled and sent to EPHI's National Reference Laboratory.

On July 28, 2023, four cases were confirmed, with a 36% positivity rate for dengue serotype DNV-3, prompting a dengue outbreak declaration. Rapid Response Teams (RRT) consisted a surveillance officer, clinician, laboratory technician, and risk communication officer who work together to tackle outbreaks. Their responsibilities include investigating the outbreaks, assisting local health personnel in implementing control measures, and evaluating the effectiveness of strategies. A taskforce, formed with local authorities and community leaders, aims to raise awareness and mobilize resources. This approach ensures that responses are coordinated and tailored, leading to more effective management of dengue outbreaks.

The team conducted a retrospective investigation of the index case, and a 12-year-old female patient was identified as the index case on July 25, 2023. They interviewed her and her relatives to evaluate her activities and potential exposures two weeks before symptom onset, mapping her residence, workplace, and other visited locations for mosquito breeding sites. She had traveled from the Afar region of Ethiopia, where dengue is endemic, to visit relatives in Metahara Town, of Dire Gobu Kebele. The team then surveyed mosquitoes and their larvae around her relatives' home and other areas.

They identified *Aedes aegypti* larvae in open water containers stored for over three days, indicating a breeding site for the primary dengue vector. The investigators identified breeding sites for *Aedes aegypti*, the primary dengue virus vector. In response, the team initiated vector control measures, including larviciding, source reduction, and community clean-up campaigns aimed at eliminating *Aedes aegypti* breeding grounds and preventing the further spread of the outbreak.

**Outbreak description.** According to the epidemic (EPi) curve, the outbreak began on July 25, 2023, marked by the index case. The official start of the dengue outbreak was recorded on July 28, 2023. The curve's shape indicates a propagative outbreak lasting 37 days. The first

peak occurred between July 31 and August 4, 2023; the second peak from August 6 to August 11, 2023; the third from August 13 to August 23, 2023; and the final peak between August 26 and September 5, 2023.

The curve suggested that initial exposure likely occurred between July 21 and 23, 2023. Conversely, the probable start date of the outbreak was estimated to be between July 27 and July 29, 2023. We emphasized that these are estimates; the actual dates of initial exposure and outbreak onset may have occurred earlier due to factors such as the completeness of case reporting, the sensitivity of the surveillance system, and the possibility of undetected or asymptomatic cases. (See Fig 2)

From July 25 to September 5, 2023, there were a total of 342 reported dengue cases, resulting in three deaths, which corresponds to a case fatality rate of approximately 0.88% to 1%. The overall attack rate (AR) during this period was 7.1 per 1,000 individuals. Out of all reported cases, 6 (1.8%) were managed as inpatients, while 336 (98.2%) received outpatient treatment. Among the cases, 182 (56%) were male, and the age group most affected was 25-44 years, comprising 189(55%) individuals (see Table 1).

Among the signs and symptoms reported by patients, fever and headache were present in 100% of cases, making them the most common indicators, while rash was the least frequently observed symptom. Of the 342 dengue cases documented, 24 (7%) progressed to severe dengue. In Metahara Town, Village Five and Village Six each accounted for 96 cases (28%), representing the highest caseloads, whereas Village Four had the lowest with only 7 cases (2%).

**Environmental investigation.** From August 3 to August 20, 2023, the investigation team conducted a thorough assessment of mosquito breeding sites by visiting households. They

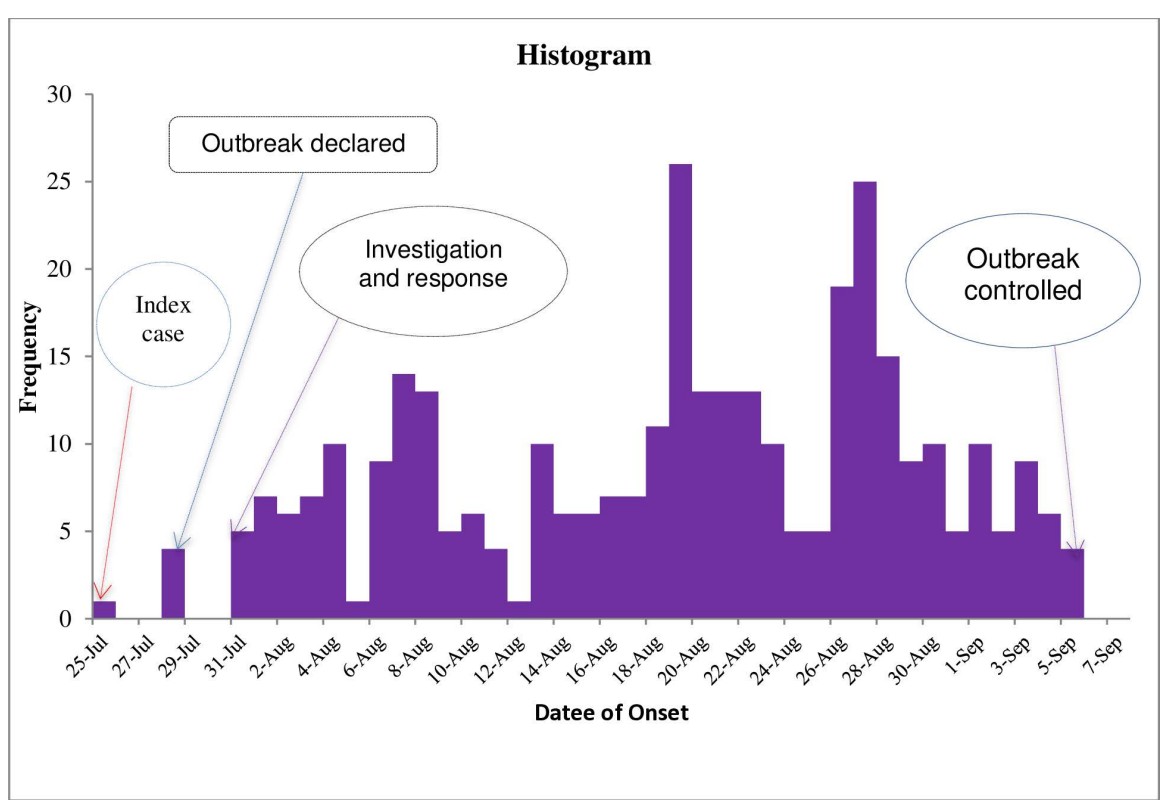

**Fig 2. Dengue outbreak epicurve.**

**Table 1. Age and Sex Category of Dengue Cases in Metahara town, East Shewa, Oromia, Ethiopia, 2023.**

| Age cat. | Population at risk | | | Sex category | | | Sex specific attack rate/1000 | | Overall attack rate/1000 |
|---|---|---|---|---|---|---|---|---|---|
| | M | F | Tot | M | F | Tot | M | F | |
| 0-14 | 9544 | 9430 | 18974 | 32 | 29 | 61 | 3.35 | 3.08 | 3.21 |
| 15-24 | 4631 | 4649 | 9280 | 36 | 40 | 76 | 7.77 | 8.60 | 8.19 |
| 25-54 | 7798 | 7892 | 15690 | 105 | 84 | 189 | 13.47 | 10.64 | 12.05 |
| 55-64 | 93 | 2014 | 2107 | 6 | 3 | 9 | 64.44 | 1.49 | 4.27 |
| +65 | 54 | 1556 | 1611 | 3 | 4 | 7 | 55.10 | 2.57 | 4.35 |
| Total | 22120 | 25541 | 47661 | 182 | 160 | 342 | 8.23 | 6.26 | 7.18 |

evaluated sanitation practices, the presence of stagnant water, water collection methods, drainage systems, individual mosquito protection measures, and the existence of uncovered water containers, window screens, and empty containers both inside and outside homes. During our home-to-home visits, we identified a total of 2,219 potential mosquito breeding sites across 1,789 households. Of these containers, 1,947 (87.7%) tested positive for dengue larvae. The investigation revealed several common mosquito breeding sites in the area, including discarded tires, plastic bottles, jerry cans, and artificial stagnant water sources frequently found in the affected communities. Due to the hot weather in Metahara Town, many residents wear short-sleeved shirts and often sleep outside during the day. The presence of the *Aedes aegypti* mosquito species in the study area was confirmed using standard morphological identification techniques [27]

## Cases-control study

**Socio-demographic characteristics.** In the case-control study, 50 cases (1:1 ratio of male to female) and 100 controls, from which 59 (59%) of the controls were males. The mean and standard deviation age of the cases and control was 38 ± 17 and 34± 12, respectively. This result indicates that a substantial majority of the participants, specifically 128 (85.3%), did not have close contact with others within the last two weeks during the study (Table 2).

**Knowledge about toward dengue.** In the case-control study, of the 50 cases, 16 (32%) reported awareness of dengue, compared to 104 (69.3%) of the controls. Only 7 individuals (14%) in the case group were aware of Aedes aegypti mosquito biting times, compared to 77 (77%) individuals in the control group.. Furthermore, 10 (20%) of the cases understood that *Aedes aegypti* mosquitoes require water for breeding, in contrast to 84 (84%) of the controls who were knowledgeable about this requirement. Additionally, a significant proportion of cases, 34 (68%), were unaware of dengue symptoms, whereas only 15 (15%) of the controls lacked this knowledge. These findings underscore a notable disparity in dengue awareness and knowledge between cases and controls, highlighting the need for targeted educational interventions to enhance understanding and prevention strategies within affected communities.

**Risk factors associated with dengue outbreak.** The final model incorporated several factors related to dengue risk, including the availability of Long-Lasting Insecticide Nets (LLINs), daytime use of LLINs, the presence of water-holding containers around the home, travel history over the past two weeks, clothing habits, and awareness of dengue. Individuals who did not use Long-Lasting Insecticide Nets (LLINs) during daytime were 9.07 times more likely risk of developing Dengue (AOR= 9.07, 95% CI: 2.05-40.06) compared to those who use it. Similarly, households with water containers in or around their premises had a 4.99-fold increased likelihood of dengue infection (AOR = 4.99, 95% CI: 21.75-14.23) compared

**Table 2. Demographic characteristics of Dengue cases and control in Metahara town, 2023.**

| Variables | Categories | Control | Case |
|---|---|---|---|
| Sex | Female | 41(41) | 25(50) |
| | Male | 59(59) | 25(50) |
| Age category | <=4 | 0(0) | 1(2) |
| | 5-14 | 0(0) | 3(6) |
| | 15-34 | 62(62) | 19(38) |
| | 35-44 | 19(19) | 10(20) |
| | >=45 | 19(19) | 17(34) |
| Marital Status | Single | 26(26) | 11(22) |
| | Married | 74(74) | 39(78) |
| Educational status | NA | 0(0) | 2(4) |
| | Unable to read and write | 5(5) | 10(20) |
| | Primary | 8(8) | 14(28) |
| | Secondary | 62(62) | 11(22) |
| | College and above | 25(25) | 13(26) |
| Occupational status | Employed | 19(19) | 13(26) |
| | Merchant | 39(19) | 14(28.2) |
| | Daily laborer | 23(23) | 16(33.3) |
| | Students | 19(19) | 5(10.4) |
| Close contact within the last 1 to 2 weeks | No | 85(85) | 43(86) |
| | Yes | 15(15) | 7(14) |

to those without such containers. Moreover, respondents who had not heard about dengue were 25.2 times more likely to contract the infection (AOR = 25.2, 95% CI: 8.21-77.34) than those who were informed. Conversely, individuals wearing long-sleeved clothing were 75% less likely to develop dengue (AOR = 0.25, 95% CI: 0.08-0.76) compared to those wearing short-sleeved attire (Table 3). These findings highlight critical behavioral and environmental risk factors associated with dengue transmission, emphasizing the importance of public health interventions aimed at increasing awareness and promoting protective measures within communities at risk (see Table 3).

## Response activities to dengue outbreak

**Coordination.** In accordance with national guidelines that classify a single confirmed case of dengue as an outbreak, a dengue outbreak was officially declared in Metahara town. This declaration followed a coordinated response involving the Metahara Health Office, Oromia Health Bureau, EPHI, East Shewa Zonal Health Office, and WHO, initiated on July 28, 2023. A comprehensive dengue response plan was developed and disseminated among stakeholders, resulting in the formation of an eight-member task force and a five-member Rapid Response Team (RRT). Daily meetings with the RRT and bi-weekly reviews with the task force have been systematically conducted to ensure effective management of the outbreak.

**Surveillance activities.** On July 29, 2023, virtual orientations were conducted, followed by two days of training for 24 health workers from Metahara town's urban health extensions and 25 from Fantale district, including personnel from PHEM offices and district health office management, totaling 49 health workers trained face-to-face from August 1 to 2, 2023. From August 3 to 5, onsite orientations were provided to staff at seven medium clinics and Marti Hospital regarding the initiation of line listing and reporting protocols to the Metahara town PHEM officer, as well as precautions for case management and integration

**Table 3. Bivariate and multivariate analysis of risk factors toward dengue outbreak in Metahara town, 2023.**

| Variables | Cat. | Status | | COR(95% CI) | AOR(95% CI) |
|---|---|---|---|---|---|
| | | Control | Case | | |
| Availability LLINs | No | 4(4%) | 7(14%) | 3.91(1.09-14.05) | 5.45(0.78-38.21) |
| | Yes | 96(96%) | 43(86%) | | |
| LLINs use on daytime | No | 70(70%) | 47(94) | 6.71(1.94-23.27) | 9.07(2.05-40.06)** |
| | Yes | 30(30%) | 3(6%) | 1 | 1 |
| Presence of water container | No | 60(60%) | 15(30%) | 1 | 1 |
| | Yes | 40(40%) | 35(70%) | 3.5(1.7-7.23) | 4.99(1.75-14.23)** |
| Travel history | No | 75(75%) | 36(72%) | 1 | 1 |
| | Yes | 25(25%) | 14(28%) | 1.17(0.54-2.51) | 1.18(0.31-4.45) |
| Wearing long Sleeve | No | 40(40%) | 32(64%) | 1 | 1 |
| | Yes | 60(60%) | 18(36%) | 0.28(0.19-0.76) | 0.25(0.08-0.76)* |
| Hear about dengue | No | 88(88% | 16(32%) | 15.58(6.68-36.34) | 25.2(8.21-77.34)** |
| | Yes | 12(12%) | 34(68%) | 1 | 1 |

with health extension services for active case searches. Surveillance tools, including line lists, case definitions, and formats for mass campaigns and health education, were printed and distributed, initiating active surveillance from August 5, 2023, until the outbreak's conclusion. Of the 150 febrile cases identified and referred to health centers, 54 were confirmed as dengue cases. An entomological investigation of the vectors was also conducted.

**Vectors control activities.** From August 5 to 10, 2023, a total of 20 liters of Abet Chemical, a larvicidal, was distributed and applied to identified breeding sites of *Aedes aegypti* mosquitoes. A comprehensive survey revealed 2,219 breeding sites, with 2,083 (93.9%) testing positive for *Aedes aegypti* larvae. Source reduction activities were conducted at 1,947 of these sites, while larvicidal spraying was performed at 136 locations. This targeted approach aimed to mitigate mosquito populations and interrupt the transmission cycle of the dengue virus by reducing larval habitats and enhancing vector control efforts.

**Diagnosis and case management.** Dengue case management activities, which focused on supportive care and early intervention to improve patient outcomes, were conducted. Key activities included fluid replacement therapy, close monitoring of vital signs, timely laboratory testing, training healthcare providers, and availing necessary supplies, which resulted in reduced case-fatality rates. Enhanced adherence to evidence-based protocols has been shown to improve patient outcomes. Overall, these management strategies emphasized the importance of early diagnosis, effective supportive care, and ongoing training for healthcare professionals to mitigate the impact of dengue outbreaks..

**Risk communications and community engagement (RCCE).** Beginning August 5, 2023, the team implemented a community health education initiative aimed at enhancing awareness and prevention of Dengue. Leaflets were prepared in local languages and distributed to community members. Health education sessions were conducted using loudspeakers at public gathering points to reach a wider audience. Sensitization efforts were organized across all kebeles in collaboration with the Ethiopian Public Health Institute (EPHI), the Oromia Health Bureau, and the task force. Additionally, district and zonal social media platforms, along with FM radio broadcasts, were utilized for effective risk communication. This multifaceted approach underscores the importance of community engagement in dengue prevention efforts, aligning with evidence that targeted health education can significantly improve knowledge and behaviors related to disease prevention.

### Challenges during outbreak response

The Oromia region is currently facing multiple outbreaks, including cholera and malaria, coupled with a shortage of trained personnel in arboviral diseases. There is an absence of a governmental contingency budget specifically allocated for arboviral-related diseases, and no partners have established planned activities or budgets for the prevention and control of these diseases in the area. In response to the outbreaks, engagement from health workers has been inadequate, compounded by a shortage of trained personnel for vector control and irregular rainfall patterns that affect mosquito populations. Additionally, high population mobility presents further challenges to effective outbreak management. To address these issues, it is essential to integrate outbreak response efforts and develop comprehensive emergency response plans that are shared with relevant authorities. Daily experience sharing and feedback mechanisms have been implemented as a supportive strategy to navigate these challenges effectively. This multifaceted approach aims to enhance the region's preparedness and response capabilities for arboviral outbreaks.

## Discussion

Dengue has emerged as a significant public health concern in Ethiopia due to several factors, including inadequate arboviral surveillance, challenges in confirming cases, and ineffective strategies for isolating infected individuals from the general population. These issues stem from limited healthcare resources, insufficient public awareness of the disease, and poor implementation of vector control measures. Current surveillance efforts have revealed substantial gaps in knowledge regarding transmission dynamics, particularly among vulnerable populations. Addressing these challenges is crucial for improving public health responses and mitigating the impact of dengue in Ethiopia. The objective of this study was to investigate the magnitude of the dengue outbreak in Metahara and identify associated factors critical for effective prevention, control, and treatment. Understanding the current burden of dengue and its risk factors is crucial for preventing future outbreaks and other arboviral diseases in Ethiopia.

This study specifically addresses the dengue outbreak in Metahara town, Oromia Region, which was confirmed through laboratory testing and represents the first documented case of dengue in this Oromia Region State. Prior to this, the only laboratory-confirmed dengue outbreak in Ethiopia was in Dire Dawa in 2013. Subsequent suspected cases have been reported across various regions of Ethiopia, including the Afar Region in 2014 and the Somali Region from January to June 2017. These findings highlight the urgent need for improved surveillance and response strategies to address the growing threat of dengue in Ethiopia [7,25,26,28].

All dengue patients in this study exhibited fever and headache, consistent with prior findings from Dire Dawa. This observation is consistent with the results of a previous investigation conducted by the city government of Dire Dawa, Ethiopia, which also reported similar symptomatology in dengue patients. These findings underscore the critical need for enhanced surveillance and diagnostic capabilities to accurately identify and manage dengue cases within the region [25] as well as an investigation of the dengue outbreak in Warder Town, Dollo Zone, Somali Region, Ethiopia [26].

The current investigation into the dengue outbreak indicates a higher prevalence among male populations compared to females, consistent with findings from a study conducted in Dire Dawa in 2017, which also reported greater susceptibility in males [29]. The age group most affected was 25-44 years, comprising 189 individuals (55%). However, this proportion is lower than that observed in a study from Warder Town, Dollo Zone, Somali Region, Ethiopia, in 2022, where 62% of individuals aged 25-44 were affected by the dengue outbreak [28].

Case fatality rates varied across regions, with Ethiopia's rate of 0.88% aligning with Bhutan and Nepal (~1%) but significantly lower than India, Indonesia, and Myanmar (3-5%),while Thailand reported a lower rate of less than 0.2% [30]. These findings highlight the need for targeted public health interventions and improved surveillance systems to manage and mitigate the impact of dengue outbreaks effectively. The observed discrepancy in dengue infection rates may be attributed to variations in the personal characteristics of study participants, differences in study design, and geographical contexts.

In the multivariate analysis, it was established that dengue patients who did not utilize long-lasting insecticide-treated nets (LLINs) during the daytime were 9.07 times more likely to develop dengue. This finding aligns with a study conducted in Dire Dawa Administration City, Ethiopia, which identified the non-use of bed nets as an independent risk factor for dengue infection [25]. These results highlight the critical role of vector control measures, particularly the use of LLINs during daytime, in reducing the risk of dengue transmission.

The current investigation indicates that households with water containers in or around their homes are 4.99 times more likely to be affected by dengue compared to those without such containers. Studies have consistently shown that the presence of stagnant water significantly elevates the likelihood of dengue outbreaks, as these environments provide ideal breeding grounds for mosquito populations. For instance, a case-control study in Dire Dawa revealed that individuals living near stagnant water were at a heightened risk of contracting Dengue, with an adjusted odds ratio (AOR) of 3.61 [7]. This aligns with broader epidemiological trends observed in other regions, where proximity to open water sources has been linked to increased morbidity associated with dengue.

The findings of this study indicate that respondents who had not heard about dengue cases were 25.2 times more likely to be infected compared to those who possessed information about the disease. This significant association underscores the critical role of awareness and education in preventing dengue infections. This observation is consistent with a study conducted in Tanzania, which similarly highlighted the correlation between knowledge of dengue and infection rates [31] and in Dollo town, Somali Region, Ethiopia [28].

The current study reveals that respondents who wore long-sleeved clothing were 75% more likely to protect themselves against dengue infection compared to those who wore short sleeves. This finding aligns with a case-control study conducted during the dengue outbreak in Warder Town, Dollo Zone, Somali Region, Ethiopia, in 2022, which indicated that wearing long sleeves provided a 43.5% greater level of protection than wearing short sleeves [26]. The protective effect of long-sleeved clothing can be attributed to its ability to reduce skin exposure, thereby minimizing the likelihood of mosquito bites. *Aedes aegypti*, the primary vector responsible for dengue transmission, is most active during the daytime, making individuals particularly vulnerable if they are inadequately clothed [10].

## Limitation

Community surveys assessing preventive practices and awareness may be susceptible to recall bias, wherein participants might inaccurately remember their past behaviors or knowledge regarding dengue prevention measures. This limitation could affect the reliability of the data collected, potentially skewing the results and interpretations of the study. The study may not have comprehensively accounted for all environmental factors that influence dengue transmission dynamics. Variables such as microclimatic conditions or unmeasured socio-economic factors could significantly impact vector breeding sites and human exposure to dengue-carrying mosquitoes, thereby affecting the overall findings.

The lack of longitudinal data limits the assessment of dengue trends and the evaluation of long-term control measures. Without such data, it is challenging to determine causality or the

sustainability of interventions over time. The effectiveness of community education initiatives can vary significantly based on local engagement levels, which may not have been adequately measured in this study. Community involvement is key to evaluating the success of educational campaigns against dengue. Discrepancies between reported practices and the actual implementation of recommended vector control measures may exist within the community.

This gap can lead to inconsistencies between intended and actual outcomes, complicating the assessment of intervention effectiveness. These limitations highlight the necessity for further research employing robust methodologies, including longitudinal studies and larger sample sizes, to enhance the understanding of dengue transmission dynamics and inform effective public health strategies in Metahara town.

## Conclusion

The transmission dynamics of Dengue in Metahara town are characterized by a complex interplay of epidemiological, entomological, and environmental factors that significantly influence outbreak patterns. Recent outbreaks have underscored the critical role of *Aedes aegypti* as the primary vector for the disease. Elevated larval indices in the area indicate insufficient vector control measures, which exacerbate the risk of transmission. The ongoing circulation of dengue serotype DENV-3 in Metahara is particularly concerning as it is associated with severe clinical manifestations, including dengue hemorrhagic fever (DHF).

The study identified several factors contributing to the dengue outbreaks: The absence of LLINs during daytime hours leaves individuals vulnerable to mosquito bites when *Aedes aegypti* mosquitoes are most active. The accumulation of standing water around homes provides ideal breeding sites for *Aedes aegypti* mosquitoes, facilitating increased vector populations. Many residents lack essential information about dengue transmission and prevention, reducing their ability to protect themselves and inadequate protective clothing choices contribute to increased exposure to mosquito bites.

Public health authorities must prioritize implementing Integrated Vector Management and community education to address key risk factors identified in this study.

## Supporting information

**S1 Text.  Consent Form.**
(DOCX)

**S2 Text.  Dengue outbreak Investigation tool.**
(DOCX)

## Acknowledgments

We express our gratitude to Jimma University Institute of Health, Faculty of Public Health, for permitting us to conduct this research. We appreciate the guidance from our academic mentor and the FETP Resident Advisor at Jimma University. We thank the field supervisor and field mentor at the Oromia Health Bureau for their support. Our heartfelt thanks extend to the Centers for Disease Control and Prevention and the Ethiopian Public Health Institute for their contributions. Finally, we acknowledge all staff members of the Oromia Health Bureau, the Metahara town administration, and health personnel who provided essential guidance during this research

## Author contributions

**Conceptualization:** Bikila Negesa Gobena.

**Data curation:** Bikila Negesa Gobena.

**Formal analysis:** Bikila Negesa Gobena, Teshome Kabeta Dadi, Gemechu Chemeda Feyisa, Dabesa Gobena.

**Funding acquisition:** Bikila Negesa Gobena.

**Investigation:** Bikila Negesa Gobena, Birhanu Kenate, Gemechu Shumi, Fantahun Workie, Haimanot Workie, Ebise Djirata.

**Methodology:** Bikila NegesA Gobena.

**Project administration:** Bikila Negesa Gobena, Birhanu Kenate, Gemechu Shumi, Fantahun Workie, Haimanot Workie, Ebise Djirata.

**Resources:** Bikila Negesa Gobena.

**Software:** BikilA Negesa Gobena.

**Supervision:** BikilA Negesa Gobena, Teshome Kabeta Dadi, Birhanu Kenate, Gemechu Shumi.

**Validation:** Bikila Negesa Gobena, Teshome Kabeta Dadi, Gemechu Chemeda Feyisa, Dabesa Gobena.

**Visualization:** Bikila Negesa Gobena.

**Writing – original draft:** Bikila Negesa Gobena, Gemechu Chemeda Feyisa, Dabesa Gobena.

**Writing – review & editing:** Bikila Negesa Gobena, Teshome Kabeta Dadi, Gemechu Chemeda Feyisa, Dabesa Gobena.

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
