## [Decision Letter · Decision Letter 0]

30 May 2024

Dear Mr. GOBENA,

Thank you very much for submitting your manuscript "Dengue Fever Outbreak Investigation and Associated factors in Metahara town, East Shewa, Oromia, Ethiopia, 2023" for consideration at PLOS Neglected Tropical Diseases. As with all papers reviewed by the journal, your manuscript was reviewed by members of the editorial board and by several independent reviewers. In light of the reviews (below this email), we would like to invite the resubmission of a significantly-revised version that takes into account the reviewers' comments. 

We would like to thank all reviewers for their expertise and time in assessing this ms. 

This is an important region for dengue virus outbreaks as Reviewer 3 has raised and a competent investigation, however a large number of revisions are requested. Please address all comments raised by the Reviwers 1 to 3 and the AE which are listed below.

Please note Reviewer 3 has provided a large number of additional comments as an attachment. Please contact the journal if the attachment is not available for download (this has happened previously) and the comments will be supplied by email. 

We cannot make any decision about publication until we have seen the revised manuscript and your response to the reviewers' comments. Your revised manuscript is also likely to be sent to reviewers for further evaluation.

Sincerely,

Michael W Gaunt, PhD

Academic Editor

Nigel Beebe

Section Editor

# Comments from AE

This is an interesting study but there is a lot of areas for improvement prior publication and is the conclusion of the majority of reviewers.

* Please supply line numbers throughout the manuscript.

* There are a large number of grammatical errors in the manuscript, some are listed by the reviewers, please have the manuscript independently proofed.

For example, "Dengue fever patients those did not use LLINs during day time were 9.07 time more likely to had odd of developing dengue fever (AOR= 9.07, 95% CI: 2.05-40.06)"

Second example, "Twenty liter of abate® chemical was given for distributed for town health office"

Third example, "regional state, Ethiopia The "

There are many examples across the ms.

* The Excel graphs are below publication standard, for example Fig. 2 the title of the x-axis mixes with the labels.

* The tables are not publication format and the formats keep changing between tables

* CAR and ASAR need defining (Table 1)

* Table 2, this is fine for the appendix and some demographic characteristics. Age distribution needs plotting between total population and case control to assess the distributions.

* Table 2, I personally suggest removing the percentages - it becomes difficult to read. It is debacle whether other demographics are better plotted particularly Occupation and Education.

*It appears adjusted odds ratio analysis did not include, gender martial status, education, occupation and age. Why not?

* LLIN AOR this makes absolute sense, but there is very little background information on a key result. The Introduction and Conclusion needs to go into substantial more detail about LLIN day time use. How does this work? How is it linked to profession? How do people practical use an insecticide net during the day? How do you conclusions link to future improved disease prevention?

* The entire discussion should be expanded.

* Please re-write your Abstract: a) to address the large number of typos therein, b) help us understand the result in an epidemiological context (e.g. insecticide nets) and b) generally the results of AOR and other tests are presented in words in an abstract state, e.g. whether a given result is significant (in words) . For example AOR is never defined in the Abstract, so it's really confusing what it means for anyone outside epidemiology. d) the conclusion needs strengthening, e) none of results are discussed in the conclusion, for example issue recommendations for control.

* Abstract and analysis "Male and age between 15-44 years old were more affected. " needs statically verifying and stating in writing.

I will be reviewing all statistics on resubmission.

Reviewer's Responses to Questions

**Key Review Criteria Required for Acceptance?**

**Methods**

-Are the objectives of the study clearly articulated with a clear testable hypothesis stated?

-Is the study design appropriate to address the stated objectives?

-Is the population clearly described and appropriate for the hypothesis being tested?

-Is the sample size sufficient to ensure adequate power to address the hypothesis being tested?

-Were correct statistical analysis used to support conclusions?

-Are there concerns about ethical or regulatory requirements being met?

Reviewer #1: -Are the objectives of the study clearly articulated with a clear testable hypothesis stated?

Yes

-Is the study design appropriate to address the stated objectives?

Yes

-Is the population clearly described and appropriate for the hypothesis being tested?

Yes

-Is the sample size sufficient to ensure adequate power to address the hypothesis being tested?

Yes

-Were correct statistical analysis used to support conclusions?

No

-Are there concerns about ethical or regulatory requirements being met?

No

Reviewer #2: Page 4, paragraph 2: I suggest changing “September2023” to “September 2023.”

Page 4, paragraph 3: I suggest changing “The study design was descriptive study” to “Study design: A Descriptive study of a dengue outbreak was conducted followed…”.

Page 4, paragraph 4, sample size section: the authors referred to “…odds ratio of 0.31 for no close contact in the previous two weeks…”. Also, they said, “control sample people of Metahara town who were neighbors to a case but did not develop signs and symptoms of DF were included as controls”. In addition, the variable close contact was not shown in the results. I suggest clarifying the definition of close contact and showing the results of the analysis of this variable. 

Page 5, paragraph 2, sampling procedure section: I suggest specifying the sampling procedure for controls (sampling was in the same block? Or 250 mt around?). In addition, the authors said, “the considering dengue fever incubation time after dengue fever cases were discontinued.” I suggest specifying the time.

Page 5, paragraph 3, The eligibility criteria section: I suggest changing “The eligibility criteria were as follows;” to “Eligibility criteria: (1)…”. In addition, considering that controls were selected in case’s neighbor, I suggest changing “(2) All case/ controls who were not residents of Metahara town” to “(2) All case who were not residents of Metahara town.”

Page 5, paragraph 5, Data collection procedures section: “The data was collected through face to face interview using questionnaire, Health facilities report by using line list, laboratory test result, Environmental investigation, and document review”. This information was repeated in the same paragraph. I suggest eliminating it. 

Page 6, paragraph 1, Definition terms: I suggest moving this information to “Eligibility criteria” section. In addition, I suggest clarifying the definition of confirmed dengue case because in sample size section the authors said, “it confirmed with Real-Time Polymerase Chain Reverse transcriptase (RT-PCR) or Rapid Diagnostic Test (RDT)…” and in “confirmed cases” did not mention RT-PCR. Also, it is important to clarify de definition of “epidemiologically connected” (time and space definition).

Page 6, paragraph 4: I suggest eliminating the sentences “Finally, the study result was displayed by using graph, figure and table.”

Reviewer #3: 1. Improve the clarity of Figure 1 (Map of Metahara town) by following PLOS-NTD image standards and formats, such as PNG, TIFF, or PDF with a resolution of 300 dpi. 

2. The description of sampling procedures is lacking details on how blood samples were collected from cases and control groups and transported to the testing laboratory. 

3. For case definitions: Epidemiological case: Why were positive malaria test cases excluded? The presence of malaria does not rule out concurrent dengue fever infection, particularly during outbreaks or in endemic tropical settings where malaria-dengue co-infection is common. For example, in Sudan, malaria-dengue co-infections have been recorded up to >10% in some areas (Ahmed et al., 2021) Source. Confirmed case: The definition needs to be revised to follow the WHO dengue guidelines for diagnosis, treatment, prevention, and control (WHO, 2009). IgM antibodies can persist in the blood for up to three months, making it difficult to differentiate between active infection and recent exposures. According to WHO, a confirmed case is RT-PCR+ within 2-7 days after the onset of fever or IgM+ seroconversion in paired sera (acute sample tested against convalescent sample collected after 7 days). The use of RT-PCR in the study should be clearly included in the confirmed case definitions. IgM+ alone does not meet the criteria for a confirmed dengue case as per WHO guidelines. 

4. The description of laboratory testing of samples is missing. Add an explanation of laboratory testing of samples using RDT and later using the RT-PCR method. 

5. The statistical analysis was correct for the intended study objectives. 

6. Ethical clearance was properly adhered to in the study.

**Results**

-Does the analysis presented match the analysis plan?

-Are the results clearly and completely presented?

-Are the figures (Tables, Images) of sufficient quality for clarity?

Reviewer #1: -Does the analysis presented match the analysis plan?

Yes

-Are the results clearly and completely presented?

Yes

-Are the figures (Tables, Images) of sufficient quality for clarity?

No

Reviewer #2: Figures and tables should be cited in the text.

Page 7, paragraph 1: I suggest changing “dengue fever serotype III” to “DENV-3”.

Page 7, paragraph 1: I suggest avoiding the mention of the name of the index case. 

Page 7, paragraph 1: The name of the mosquito must be correct throughout the text and written in Italic letters. It is “Aedes aegypti”. 

Page 7, paragraph 2: The year must be correct (2025 to 2023).

Page 8, paragraph 1: The year must be correct (2024 to 2023).

Page 8, paragraph 1: The authors mentioned, “On the other hand, the probable date of the outbreak could be between July 27 and July 29, 2023. It is important to note that this was an estimate, as the actual initial exposure and outbreak start date may be earlier or later, depending on factors such as the completeness of case reporting…”, and before they said, “the date of onset was 25th July, 2025 which was index case and in 28th, 2023 was when dengue fever outbreak was confirmed which the start of the outbreak”. Taking into account this information and the epidemic curve presented (Figure 2), I suggest changing “outbreak start date may be earlier or later” to “outbreak start date may be earlier…” because later is not possible.

Page 8, paragraph 2: According to the Table 1, the attack rate must be correct (718/10,000 for 718/100,000). 

Page 8, paragraph 2: The following sentence is incomplete “From all cases 6(1.8%) were managed as inpatient whereas 336(98.2%).” 

Page 8, paragraph 2: The following sentence is incomplete “From all cases 6(1.8%) were managed as inpatient whereas 336(98.2%).” 

Page 8, paragraph 2: The authors said, “15-44 were predominantly 189(55%) exposed group”. However, the Table 1 shows 25-54. Please clarify it. In addition, I sugest changing “exposed group” to “affected group”.

Page 9, laboratory investigation section. I suggest eliminating this section because the information was mentioned in “Dengue fever outbreak verification section”.

Page 9, paragraph 2, Environmental Investigation section: I suggest moving “ The investigation team noticed breeding site for mosquitoes from house to house, sanitation procedures, the presence of stagnant water, water collection practices, drainage systems, and individual mosquito protection measures, as well as the presence of uncovered water containers, window screening, and empty containers both inside and outside the homes” to methods. 

Table 2. Is it “Not applicable” or “not available”? In addition, I suggest changing “N” to “n”. Also, to simplify, I suggest eliminating “%” of all cells and writing it just once. 

Table 3. Please verify the CI95% in “1.18(0.31-445)”, as it looks too wide.

Page 12, Coordination section: I suggest including dates of the activities. 

Page 13, surveillance section: I suggest including dates of the activities. 

Page 13, vector control activities section: I suggest including dates of the activities. 

Page 14, Diagnosis and Case Management: I suggest eliminating “Of 342 lines listed cases 24 were diagnosed for sever dengue and two facility deaths were reported with Dengue shock syndrome”. This information was presented earlier and there is an inconsistency, as the authors previously stated that the deaths were three, but in this part, they mention two.

Page 14, Risk Communications and Community Engagement section: I suggest including dates of the activities.

Reviewer #3: 1. The analysis provided agrees with the analysis plan. 

2. The results are not clearly presented. 

3. The tables should be formatted according to standard publication rules, e.g., tables should have no internal lines except for headers. 

4. I recommend recategorizing the age groups to indicate extreme age groups, e.g., 0-4 yrs, 5-14 yrs, 15-24, 25-34, 35-44, 45-54, 55-64, 65+. This is important in the context of dengue immune response and severe disease development, which differ between these age groups, with higher risk observed in children and adults over 60. Immunity for preschool and school children may differ due to varying levels of exposure to dengue infection, with higher risk in the older group (>50) due to enhanced immunity following secondary infections with different dengue types that lead to cross-reactive immune complexes and release of inflammatory agents. These age groups should be consistent in Table 1 and Table 2. 

5. Section 3.2.3, "Factors associated with dengue fever outbreak," should be revised to "Risk factors associated with dengue fever outbreak." 

6. All figures lack captions (subtitles) and explanations. They should be well-labeled, e.g., Figure A, B, C, etc., and include the source and date, e.g., Source: John et al., 2014. 

7. There are many grammatical errors and typos in this section that need to be revised. 

8. All scientific names should be correctly written in italics, e.g., Aedes aegypti.

**Conclusions**

-Are the conclusions supported by the data presented?

-Are the limitations of analysis clearly described?

-Do the authors discuss how these data can be helpful to advance our understanding of the topic under study?

-Is public health relevance addressed?

Reviewer #1: -Are the conclusions supported by the data presented?

Yes

-Are the limitations of analysis clearly described?

Yes

-Do the authors discuss how these data can be helpful to advance our understanding of the topic under study?

Yes

-Is public health relevance addressed?

Yes

Reviewer #2: The authors said, “More cases were reported in males than females with increased severity, admission rate and case fatality rate.” However, they only presented the information on attack rate by gender. 

The authors mentioned, “Individuals aged between 15 to 44 years old were predominantly affected”. However, this age range was not presented in the table.

Reviewer #3: 1. The conclusions are supported by the data presented; however, there is significant redundant information, particularly repeated results and discussion points. For instance, results such as “more cases were reported in males than females with increased severity, admission rate, and case fatality rate” and “the presence of water containers in or around households and not using LLINs during the day were significant risk factors” are repeated without adequate implications being provided. These statements are part of the results rather than the conclusions.

2. The author should synthesize these results to provide detailed interpretations and implications. Specifically, the author should explain how these findings can inform public health interventions in Metahara town, particularly in terms of implementing or strengthening vector control, dengue surveillance, and preparedness against future outbreaks, as mentioned in the introduction section.

3. The recommendation section should be removed to adhere to the standard "IMRaDC" format; Introduction (the study's purpose), Methods (how the study was conducted and the justification for the chosen methods), Results (what was found), Discussion (the meaning of the findings), and Conclusions (implications and lessons learned from the findings).

4. The limitations of the analysis have not been discussed. Additionally, the authors have not sufficiently explained how the data advances understanding of the topic. More synthesis and interpretation are required to address this.

**Editorial and Data Presentation Modifications?**

Reviewer #1: (No Response)

Reviewer #2: In some parts of the text the old classification of dengue (dengue fever, dengue hemorrhagic) is mentioned and in others the current classification (severe dengue). Also, sometimes DF is used and other times dengue fever. Please standardize the nomenclature throughout the article.

In the discussion the results are repeated verbatim. I suggest paraphrasing these. 

Additionally, I suggest including the recommendations section within the discussion. 

I suggest checking the punctuation and the use of uppercase and lowercase letters.

Reviewer #3: A minor revision is recommended to enhance the quality of data presentation and general manuscript formatting. The paper is worthy of publication once the suggested comments have been appropriately addressed by the authors.

**Summary and General Comments**

Reviewer #1: The manuscript presents a study focused on determining a dengue outbreak in Metahara town, East Shewa, Oromia, Ethiopia. The methodology was based on obtaining blood samples and applying RT-PCR tests. The collected epidemiological data and test results identified factors associated with the dengue outbreak. After reviewing the manuscript, the following comments are highlighted:

1. The title of the manuscript is general; the authors must specify depending on the innovation of the research.

2. The references used in the introduction are insufficient. I suggest adding other works, in which different factors related to the outbreak of dengue cases are mentioned, for example:

*Mehmood, Y., Arshad, M. Dengue in the urban slums of Pakistan: health costs, adaptation practices, and the role of dengue-diagnosis and surveillance in controlling the epidemic. GeoJournal 89, 75 (2024). https://doi.org/10.1007/s10708-024-11073-y

*Conde-Gutiérrez, R.A., Colorado, D., Márquez-Nolasco, A. et al. Parallel prediction of dengue cases with different risks in Mexico using an artificial neural network model considering meteorological data. Int J Biometeorol (2024). https://doi.org/10.1007/s00484-024-02643-3

*Kumharn, W., Piwngam, W., Pilahome, O. et al. Effects of meteorological factors on dengue incidence in Bangkok city: a model for dengue prediction. Model. Earth Syst. Environ. 9, 1215–1222 (2023). https://doi.org/10.1007/s40808-022-01557-6

3. Add in the section Data processing and analysis, the related equations for the calculation of Chi-square tests, and Binary logistic regression.

4. In Figure 2, "Date of onset" is superimposed on the x-axis.

5. Present the multivariate analysis mentioned in the discussion section graphically for the greater interest of readers.

Reviewer #2: The manuscript on the study of the first dengue outbreak in Metahara, Ethiopia, and the factors associated with dengue cases is a relevant topic. The authors present detailed information about the outbreak study and the activities carried out to counteract the impact on the city. However, some aspects of the methodology must be clarified, as well as improving the presentation of the results and avoiding repetition in the text.

Reviewer #3: Ethiopia has experienced a surge in dengue virus infections since 2019, highlighting the need for research to inform public health interventions. The current study is original research focusing on Metahara town in Ethiopia, providing valuable information for public health decision-makers to improve preventive strategies and save lives. No further experiments are required in the current format, but minor revisions to address all comments are needed. Additionally, I recommend revising the manuscript title to "Dengue Fever Outbreak Investigation and Associated Risk Factors in Metahara Town, Ethiopia, 2023."

PLOS authors have the option to publish the peer review history of their article (what does this mean?). If published, this will include your full peer review and any attached files.

Reviewer #1: No

Reviewer #2: Yes: Ruth A. Martínez-Vega

Reviewer #3: Yes: Dr. Gaspary Mwanyika, Centre for Epidemic Response and Innovation of the School for Data Science and Innovation, Stellenbosch University, Stellenbosch, South Africa, Department of Applied Sciences, Mbeya University of Science and Technology, Mbeya, Tanzania
---

## [Decision Letter · Decision Letter 1]

25 Oct 2024

PNTD-D-24-00552R1Unraveling the dynamics of Dengue in Metahara town, East Shewa, Oromia, Ethiopia, 2023PLOS Neglected Tropical Diseases Dear Dr. GOBENA, Thank you for submitting your manuscript to PLOS Neglected Tropical Diseases. After careful consideration, we feel that it has merit but does not fully meet PLOS Neglected Tropical Diseases's publication criteria as it currently stands. Therefore, we invite you to submit a revised version of the manuscript that addresses the points raised during the review process. Please submit your revised manuscript within 60 days Nov 24 2024 11:59PM. If you will need more time than this to complete your revisions, please reply to this message or contact the journal office at plosntds@plos.org. Please include the following items when submitting your revised manuscript:* A rebuttal letter that responds to each point raised by the editor and reviewer(s). You should upload this letter as a separate file labeled 'Response to Reviewers'. This file does not need to include responses to any formatting updates and technical items listed in the 'Journal Requirements' section below.* A marked-up copy of your manuscript that highlights changes made to the original version. You should upload this as a separate file labeled 'Revised Manuscript with Track Changes'.* An unmarked version of your revised paper without tracked changes. You should upload this as a separate file labeled 'Manuscript'. If you would like to make changes to your financial disclosure, competing interests statement, or data availability statement, please make these updates within the submission form at the time of resubmission. Guidelines for resubmitting your figure files are available below the reviewer comments at the end of this letter. We look forward to receiving your revised manuscript. Kind regards, Michael W Gaunt, PhDAcademic EditorPLOS Neglected Tropical Diseases Nigel BeebeSection EditorPLOS Neglected Tropical Diseases

Shaden Kamhawi

co-Editor-in-Chief

Paul Brindley

co-Editor-in-Chief

 **Journal Requirements:** **Additional Editor Comments (if provided):** # Notes from the AE

The overall study is very promising. The manuscript is significantly below professional standard. The English needs an expert to review and a scientific editor. The reviewers have also raised this concern and we cannot act as unpaid editor for the authors' shortcomings.

As described by the reviewers, there are errors throughout the manuscript for example,

"Of 50, 16 (32%) cases had heard of Dengue, while 104 (69.3%) of the total controls had heard of ..."

This is not grammatical English. It is one example amongst many.

The nomenclature is erratic across the manuscript,

For example, DNV should be DENV

RT PCR should be RT-PCR

Dengue else where DENGUE. There reality is the correct spelling is dengue.

The manuscript is strewn with errors.

The additional issues are quality of the charts is very basic Excel, the GIS is not sufficient quality. The bar chart is not a good representation of the data. The tables should be converted to charts and the tables transferred to the appendix. The core analytics should be presented as a table, as the authors have done.

I am also concerned about the failure to reference many notable studies in dengue epidemiology in Africa and especially South East Asia.

I cannot let the manuscript past the editorial process until it is of sufficient standard and instruct the authors' to seek external support including expert support and scientific editing to clear up the manuscript. There is no question this an emergent infection to Ethiopia, the authors are experts, have performed a competent and this of significant public health interest.**Reviewers' Comments:** Reviewer's Responses to Questions

**Key Review Criteria Required for Acceptance?**

**Methods**

-Are the objectives of the study clearly articulated with a clear testable hypothesis stated?

-Is the study design appropriate to address the stated objectives?

-Is the population clearly described and appropriate for the hypothesis being tested?

-Is the sample size sufficient to ensure adequate power to address the hypothesis being tested?

-Were correct statistical analysis used to support conclusions?

-Are there concerns about ethical or regulatory requirements being met?

Reviewer #1: (No Response)

Reviewer #2: In general, I agree. However, please see my comments below.

**Results**

-Does the analysis presented match the analysis plan?

-Are the results clearly and completely presented?

-Are the figures (Tables, Images) of sufficient quality for clarity?

Reviewer #1: (No Response)

Reviewer #2: In general, I agree. However, please see my comments below.

**Conclusions**

-Are the conclusions supported by the data presented?

-Are the limitations of analysis clearly described?

-Do the authors discuss how these data can be helpful to advance our understanding of the topic under study?

-Is public health relevance addressed?

Reviewer #1: (No Response)

Reviewer #2: The limitations mentioned by the authors are quite general. They should be specific to the study.

**Editorial and Data Presentation Modifications?**

Reviewer #1: (No Response)

Reviewer #2: 1. It is important to conduct a thorough proofreading of the article, as there are several issues throughout the text, including misplaced full stops in sentences, inappropriate use of capital letters, punctuation errors before and after references, and instances where references are written without spaces between the preceding word and the parentheses.

2. The following text is repeted in “Study Area and Period” section. Please remove the duplicate content. “Based on the town administrative report of 2022/23, the town population was 47,661, one health center, two health posts, and 13 private clinics. The country has made significant strides in improving its disease surveillance system. Although Arboviral disease outbreaks like Yellow fever, Dengue, Chikungunya, and Sand Fly Fever in Ethiopian neighboring countries like Sudan, Eritrea, Kenya, and Djibouti, there are still limitations in enhancing arboviral disease surveillance in Ethiopia”.

3. Please correct “anti-DENVI gG/IgM” to “anti-DENV IgG/IgM”.

4. Please correct in Results, 1 paragraphac “DNV-3” to “DENV-3”.

5. Please correct “DENGUE” to “Dengue”.

6. Please correct “Arbo-virus diseases” to “arboviral diseases”

**Summary and General Comments**

Reviewer #1: (No Response)

Reviewer #2: The authors addressed most off the comments. However, there are still some that were not taken into account, and I believe they are important.

1. In the sample size section, the authors referred to an “odds ratio of 0.32 for no close contact in the previous two weeks”. They also clarified that the control group consisted of “any person who lives in same block to the patient selected as having Dengue case, but those without signs or symptoms of the disease starting from July 25 to September 5/2023”. Given this, it appears that all controls were either in relatively close contact or entirely unexposed (i.e., no close contact because they do not live in the same household as the cases). As a result, this variable becomes a constant among the controls, making impossible to calculate an OR, as there would be zero exposed control. In addition, the variable “close contact”, which was the primary exposure variable used to calculate the sample size, does not appear in the results. I recommend either clarifying the definition of “close contact” and presenting the analysis of this variable in the results or modifying the text to reflect more general information for any variable with aa similar frequency distribution.

2. The name of the mosquito should be consistent and correctly formatted throughout the text in Italic letters: Aedes aegypti. For example, in the first paragraph of the results, “Aedes Aegpti”, is misspelled, and on page 14, “anopheles Aegyptus” is incorrectly used. Please ensure that the correct species name, Aedes aegypti, is used consistently (https://wrbu.si.edu/vectorspecies/mosquitoes/aegypti ; https://www.ncbi.nlm.nih.gov/Taxonomy/Browser/wwwtax.cgi?mode=info&id=7159

3. On page 12, paragraph 1, the year should be corrected from 2024 to 2023."

4. On page 12, paragraph 2, the authors state “15-44 were predominantly 189(55%) exposed group”, but Table 1 shows the age range as 25-54. Please correct this discrepancy. The same error appears in the Discussion section on page 17.

5. Table 2. Is it “Not applicable” or “not available”?

Additional comments:

1. In “Risk factors associated with Dengue outbreak” section (page 14). I suggest changenig “Dengue patients who did not use Long-Lasting Insecticide Nets (LLINs) (LLINs) during daytime were 9.07 times more likely risk of developing Dengue (AOR= 9.07, 95% CI: 2.05-40.06) compared to those who did not use it.” to “Individuals who did not use Long-Lasting Insecticide Nets (LLINs) (LLINs) during daytime were 9.07 times more likely risk of developing Dengue (AOR= 9.07, 95% CI: 2.05-40.06) compared to those who use it”.

2. In Table 2, the suggestion was to remove the “%” symbol, not the relative frequencies. Please retain the relative frequencies, for example, 25 (50) and 41 (41).

3. I suggest moving the limitations section to the Discussion section or placing before the Conclusion section.

PLOS authors have the option to publish the peer review history of their article (what does this mean?). If published, this will include your full peer review and any attached files.

Reviewer #1: No

Reviewer #2: **Yes: **Ruth Aralí Martinez Vega

---

## [Decision Letter · Decision Letter 2]

9 Jan 2025

PNTD-D-24-00552R2Unraveling the dynamics of Dengue in Metahara town, East Shewa, Oromia, Ethiopia, 2023PLOS Neglected Tropical Diseases  Dear Dr. GOBENA, Thank you for submitting your manuscript to PLOS Neglected Tropical Diseases. After careful consideration, we feel that it has merit but does not fully meet PLOS Neglected Tropical Diseases's publication criteria as it currently stands. Therefore, we invite you to submit a revised version of the manuscript that addresses the points raised during the review process. Please submit your revised manuscript within 30 days Feb 08 2025 11:59PM. If you will need more time than this to complete your revisions, please reply to this message or contact the journal office at plosntds@plos.org. Please include the following items when submitting your revised manuscript: * A rebuttal letter that responds to each point raised by the editor and reviewer(s). You should upload this letter as a separate file labeled 'Response to Reviewers'. This file does not need to include responses to any formatting updates and technical items listed in the 'Journal Requirements' section below. * A marked-up copy of your manuscript that highlights changes made to the original version. You should upload this as a separate file labeled 'Revised Manuscript with Track Changes'. * An unmarked version of your revised paper without tracked changes. You should upload this as a separate file labeled 'Manuscript'. If you would like to make changes to your financial disclosure, competing interests statement, or data availability statement, please make these updates within the submission form at the time of resubmission. Guidelines for resubmitting your figure files are available below the reviewer comments at the end of this letter. We look forward to receiving your revised manuscript. Kind regards, Michael W Gaunt, PhDAcademic EditorPLOS Neglected Tropical Diseases Nigel BeebeSection EditorPLOS Neglected Tropical Diseases

Shaden Kamhawi

co-Editor-in-Chief

Paul Brindley

co-Editor-in-Chief

**Additional Editor Comments:** I requested a full revision of the English in the manuscript. The authors have not performed this to anything close to a satisfactory standard.

The authors are therefore requested to go through each of the 43 comments concerning grammatical and syntax errors, and issues with clarity for the Abstract, Introduction, Methods, Results and Discussion and address each in turn. On the revision state which changes have been made and give clear justification if not performed. The revision will be checked in detail to ensure all changes have been incorporated.

Abstract:

* "Dengue, a mosquito-borne viral disease, has increasingly been reported in Ethiopia, with a suspected outbreak occurring in Metahara town, Oromia Regional State, in July 2023."

Issue: The sentence is clear but could benefit from greater specificity about "increasingly been reported."

Suggestion: Specify the timeframe or trends (e.g., "since [year], dengue cases have shown a marked increase in Ethiopia").

* "This study aimed to confirm the outbreak, identify associated risk factors, describe epidemiological characteristics, and implement control measures."

Suggestion: Simplify by avoiding repetition: "This study aimed to confirm and characterize the outbreak, identify risk factors, and implement control measures."

* "A descriptive study followed by an unmatched case-control design (1:2 case-to-control ratios) was conducted."

Issue: "Followed by" is slightly unclear. Were the descriptive study and case-control study distinct phases or conducted concurrently?

Suggestion: Rephrase for clarity, e.g., "A descriptive study and an unmatched case-control study (1:2 case-to-control ratio) were conducted."

* "Data were collected using Kobo Collect and analyzed with SPSS version 26 for statistical analysis and Microsoft Excel 2013 for descriptive statistics."

Issue: Repetition of "for statistical analysis."

Suggestion: Simplify: "Data were collected using Kobo Collect, analyzed with SPSS version 26, and summarized with Microsoft Excel 2013."

"The strength of associations was measured using crude odds ratios (OR) and adjusted odds ratios (AOR), with statistical significance defined at p-values ≤ 0.05."

* Issue: Clarity is good, but the text could be condensed.

Suggestion: "Crude and adjusted odds ratios (OR and AOR) were used to measure associations, with p-values ≤ 0.05 indicating significance."

* "The investigation confirmed 342 cases of dengue, resulting in an attack rate of 7.1 per 1,000 populations, with a case fatality rate of 0.88%."

Issue: "Populations" should be singular ("population").

Suggestion: "The investigation confirmed 342 dengue cases, corresponding to an attack rate of 7.1 per 1,000 population and a case fatality rate of 0.88%."

* "Factors associated with the dengue were: not using long-lasting insecticide nets during the daytime was a ninefold increased likelihood of dengue outbreak..."

Issue: Awkward phrasing and incorrect structure.

Suggestion: "Significant risk factors included not using long-lasting insecticide nets during the daytime (9-fold increased likelihood) and having open water containers (5-fold increased likelihood)."

* "The dengue outbreak in Metahara town is significantly influenced by epidemiological, entomological, and environmental factors, with Aedes aegypti identified as the primary vector."

Suggestion: Streamline for clarity: "The dengue outbreak in Metahara town was driven by epidemiological, entomological, and environmental factors, with Aedes aegypti as the primary vector."

"Hence, strategies combining vector control measures with community education, promoting long-sleeved clothing, raising awareness about dengue transmission, and establishing robust surveillance systems were recommended."

Suggestion: Avoid redundancy: "Strategies including vector control, community education, promotion of protective clothing, and improved surveillance were recommended."

* "The Authors conducted a detailed and comprehensive investigation..."

Issue: "Authors" should not be capitalized. The phrase "detailed and comprehensive" is redundant.

Suggestion: "The authors conducted an investigation..."

* "Intermittent rains, inappropriate water storage practices, and numerous old tires in town before the outbreak had created ideal breeding grounds for these mosquitoes."

Issue: Verb tense inconsistency ("had created").

Suggestion: Use the simple past tense: "...created ideal breeding grounds for mosquitoes."

* "The outbreak spread rapidly, with hundreds of cases reported in a short period."

Suggestion: Specify the timeline if possible: "...in a span of [timeframe]."

* "A multi-faceted approach that was enhancing disease surveillance..."

Issue: Incorrect verb form.

Suggestion: "A multi-faceted approach that enhanced disease surveillance..."

* "The author's comprehensive investigation provides valuable insights..."

Issue: "Author's" is incorrect; use "authors'."

Suggestion: "The authors' investigation provides..."

Introduction, Methods and Results

* The statement "Dengue can cause asymptomatic infections as well as life-threatening complications such as dengue hemorrhagic fever and shock syndrome" is redundant as the previous sentence already conveys the same idea.

Suggested revision: Combine the sentences for clarity and conciseness.

* "Since 2013, the dengue outbreak has increased in the healthcare system of Ethiopia" could be rephrased for accuracy and clarity as "Since 2013, dengue outbreaks have increasingly burdened Ethiopia's healthcare system."

* "The first confirmed case was identified in Dire Dawa City, eastern Ethiopia ten years ago" requires a comma after "Ethiopia."

* Phrases such as "presence of anti-DENV IgG/IgM antibodies in patients who are feverish" could be simplified to "detection of anti-DENV IgG/IgM antibodies in febrile patients" for improved readability.

* The section discussing risk factors and LLIN utilization transitions abruptly. Consider reorganizing the paragraph to discuss vector control measures more cohesively.

* Some statements, like "Ethiopia has made significant progress in enhancing its disease surveillance system," lack citations for substantiation.

* The vaccine paragraph lacks clarity. Revise as: "While dengue is vaccine-preventable, vaccines like Dengvaxia and Qdenga are not yet part of Ethiopia's immunization program. The recent WHO prequalification of Qdenga may facilitate future availability."

* The description of Metahara Town's climate and geography could be streamlined for focus.

Suggested revision: "Metahara Town, in the East Shewa Zone of Oromia, Ethiopia, lies 193 km from Addis Ababa and experiences a hot, partly cloudy climate with annual temperatures ranging from 67°F to 93°F."

* "Using QGIS software, we created a map of the research area by evaluating and overlaying various data layers" is awkward. Simplify to: "We used QGIS software to create a research area map by overlaying data layers."

* Avoid passive phrases like "The sampling frame was created." Instead: "We created the sampling frame."

* "Control was any person who lives in the same block with the case" should be "A control was defined as any person living in the same block as a case."

* Specify the validity of the STANDARD Q Dengue NS1 Ag test for local conditions to enhance credibility.

* Replace "4 ml of serum during the acute phase of the disease (0–7 day’s post-symptom onset)" with "4 mL of serum during the acute phase (0–7 days post-symptom onset)."

* Combine eligibility and exclusion criteria for better organization.

* "All cases who were not residents of Metahara town" should be "All non-resident cases of Metahara Town."

* The phrase "mandated for the outbreak" is unclear. Replace with "required for outbreak investigation."

* State explicitly whether verbal consent was sufficient under the ethical clearance.

* "Primary data collection was performed using face-to-face interviews via the Kobo Collect mobile application" could be simplified as: "We collected primary data using face-to-face interviews through the Kobo Collect mobile app."

* The description of the questionnaire's components is lengthy. Use bullet points for clarity:

* Use consistent phrasing for definitions, e.g., "An epidemiological case is defined as..."

* In "Close contact is defined as being within a distance of 1 meter," change "is defined" to "was defined" to maintain past tense consistency.

* Replace "Data was checked for completeness, coded, cleaned" with "Data were checked for completeness, coded, and cleaned."

* The statistical methods need simplification for broader readability. For instance, "We used binary logistic regression to analyze the data and determine associations."

* The phrase "lifestyle modification practices" is vague. Specify the practices under discussion.

* "All testing negative for malaria", suggestion, "All of which tested negative for malaria."

* "Task forces and rapid response teams were established to investigate the outbreak's extent and implement control measures.", suggestion, Consider specifying the composition or role of these teams for added clarity.

* "According to the EPi curve", suggestion, "According to the epidemic (EPi) curve." (Always expand abbreviations on first use.)

* "The curve indicated that the potential initial exposure date could have occurred...", suggestion, Rephrase to: "The curve suggested that initial exposure likely occurred..."

* "The presence of the Aedes aegypti mosquito species in the study area was confirmed using standard morphological identification techniques[22]", suggestion, Add a period after the citation number.

* "During their visits to 1,789 households, the team identified 2,219 water-holding containers that had the potential to serve as mosquito breeding sites.", suggestion, Simplify: "...identified 2,219 potential mosquito breeding sites in 1,789 households."

* "This result indicates that a substantial majority of the participants, specifically 128(85.3%) did not have close contact...", suggestion, Add a comma after 128(85.3%) for clarity.

* Ensure consistent formatting of percentages and numbers across the text (e.g., "128(85.3%)", suggestion, "128 (85.3%).").

* Include context for why close contact and educational status are relevant to the study outcomes.

* "Knowledge regarding the biting times of Aedes aegypti mosquitoes was limited...", suggestion, Avoid repetition of "knowledge" in consecutive sentences.

* "Individuals who did not use Long-Lasting Insecticide Nets (LLINs) (LLINs)...", suggestion, Remove duplicate reference to LLINs.

* "A dengue outbreak has been officially declared in Metahara town.", suggestion, Should be past tense: "was officially declared."

* "Among 150 febrile cases identified through active case searches sent to health centers, 54 were confirmed as dengue cases.", suggestion, Clarify: "Of the 150 febrile cases identified and referred to health centers, 54 were confirmed as dengue cases."

* "This targeted approach aims to mitigate mosquito populations...", suggestion, Use past tense: "This targeted approach aimed to..."

* "The Oromia region is currently facing multiple outbreaks, including cholera and malaria, alongside a significant lack of trained personnel knowledgeable about arboviral diseases.", suggestion, "...cholera and malaria, coupled with a shortage of trained personnel in arboviral diseases."

Discussion

* "Dengue has emerged as a significant public health concern in Ethiopia, characterized by underreported epidemiology due to inadequate arboviral surveillance, challenges in case confirmation, and a lack of isolation in endemic areas."

Issues: "Underreported epidemiology" is unclear and awkward.

* The phrase "a lack of isolation in endemic areas" could be clarified.

Suggestion: "Dengue has emerged as a significant public health concern in Ethiopia due to inadequate arboviral surveillance, challenges in case confirmation, and limited isolation measures in endemic areas."

* "Understanding the current burden of dengue and identifying major risk factors are essential for mitigating future outbreaks of dengue and other arboviral diseases in Ethiopia through appropriate recommendations."

Issues: The sentence is lengthy and can be simplified.

Suggestion: "Understanding the current burden of dengue and its risk factors is crucial for preventing future outbreaks and other arboviral diseases in Ethiopia."

* "Prior to this, the only laboratory-confirmed dengue outbreak in Ethiopia occurred in Dire Dawa city in 2013."

Issues: "City" is redundant as Dire Dawa is commonly recognized as a city.

Suggestion: "Prior to this, the only laboratory-confirmed dengue outbreak in Ethiopia was in Dire Dawa in 2013."

* "The findings of this study reveal that all patients diagnosed with dengue exhibited fever and headache, with both symptoms occurring in 100% of cases among individuals presenting with dengue-related clinical manifestations."

Issues: "With both symptoms occurring in 100% of cases" is repetitive.

Suggestion: "All dengue patients in this study exhibited fever and headache, consistent with prior findings from Dire Dawa."

* Surveillance gaps and a lack of community awareness are mentioned multiple times. Consider consolidating these themes for better readability.

Suggestion: Combine mentions of the need for improved surveillance into one cohesive paragraph.

* When comparing case fatality rates across regions, the discussion could benefit from clearer transitions and summarization:

Suggestion: "Case fatality rates varied across regions, with Ethiopia's rate of 0.88% aligning with Bhutan and Nepal (~1%) but significantly lower than India, Indonesia, and Myanmar (3-5%). These differences highlight geographical and systemic variations in dengue management."

* "The absence of longitudinal data restricts the ability to assess temporal trends in dengue incidence or evaluate the long-term effectiveness of implemented control measures."

Issue: Sentence is correct but slightly long.

Suggestion: "The lack of longitudinal data limits the assessment of dengue trends and the evaluation of long-term control measures."

* "Understanding the degree of community involvement is crucial for evaluating the success of educational campaigns aimed at reducing dengue transmission."

Issue: Could be more concise.

Suggestion: "Community involvement is key to evaluating the success of educational campaigns against dengue."

* "The ongoing circulation of dengue serotype DENV-3 in Metahara is particularly concerning; as this serotype is associated with severe clinical manifestations, including dengue hemorrhagic fever (DHF)."

Issue: Misuse of semicolon.

Suggestion: "The ongoing circulation of dengue serotype DENV-3 in Metahara is particularly concerning, as it is associated with severe clinical manifestations, including dengue hemorrhagic fever (DHF)."

* "Many residents lack essential information about dengue virus transmission and prevention strategies, which diminishes their ability to protect them effectively."

Issue: "Protect them" is awkward.

Suggestion: "Many residents lack essential information about dengue transmission and prevention, reducing their ability to protect themselves."

* The conclusion could explicitly link study findings to actionable recommendations. For example:

"Public health authorities must prioritize implementing Integrated Vector Management and community education to address key risk factors identified in this study."**Journal Requirements:**

1) We note that your Data Availability Statement is currently as follows: "Data is available at the hand of principal investigator and can access if required". Please confirm at this time whether or not your submission contains all raw data required to replicate the results of your study. Authors must share the “minimal data set” for their submission. PLOS defines the minimal data set to consist of the data required to replicate all study findings reported in the article, as well as related metadata and methods (https://journals.plos.org/plosone/s/data-availability#loc-minimal-data-set-definition).

- The points extracted from images for analysis..

**Reviewers' comments:** Reviewer's Responses to Questions

**Key Review Criteria Required for Acceptance?**

**Methods**

-Are the objectives of the study clearly articulated with a clear testable hypothesis stated?

-Is the study design appropriate to address the stated objectives?

-Is the population clearly described and appropriate for the hypothesis being tested?

-Is the sample size sufficient to ensure adequate power to address the hypothesis being tested?

-Were correct statistical analysis used to support conclusions?

-Are there concerns about ethical or regulatory requirements being met?

Reviewer #2: This is the third revision. The authors have addressed the comments.

**Results**

-Does the analysis presented match the analysis plan?

-Are the results clearly and completely presented?

-Are the figures (Tables, Images) of sufficient quality for clarity?

Reviewer #2: This is the third revision. The authors have addressed the comments.

**Conclusions**

-Are the conclusions supported by the data presented?

-Are the limitations of analysis clearly described?

-Do the authors discuss how these data can be helpful to advance our understanding of the topic under study?

-Is public health relevance addressed?

Reviewer #2: This is the third revision. The authors have addressed the comments.

**Editorial and Data Presentation Modifications?**

Reviewer #2: This is the third revision. The authors have addressed the comments.

**Summary and General Comments**

Reviewer #2: This is the third revision. The authors have addressed the comments.

PLOS authors have the option to publish the peer review history of their article (what does this mean?). If published, this will include your full peer review and any attached files.

Reviewer #2: **Yes: **Ruth Aralí Martínez Vega

---

## [Editor Report · Decision Letter 3]

11 Feb 2025

Dear Mr. GOBENA,

We are pleased to inform you that your manuscript 'Unraveling the dynamics of Dengue in Metahara town, East Shewa, Oromia, Ethiopia, 2023' has been provisionally accepted for publication in PLOS Neglected Tropical Diseases.

Best regards,

Michael W Gaunt, PhD

Academic Editor

Nigel Beebe

Section Editor

Shaden Kamhawi

co-Editor-in-Chief

Paul Brindley

co-Editor-in-Chief
